# Synthesis and Insecticidal Activity of Novel Anthranilic Diamide Insecticides Containing Indane and Its Analogs

**DOI:** 10.3390/ijms25042445

**Published:** 2024-02-19

**Authors:** Zhihui Yang, Ruihan Hu, Jingjing Chen, Xiaohua Du

**Affiliations:** Catalytic Hydrogenation Research Center, Zhejiang Key Laboratory of Green Pesticides and Cleaner Production Technology, Zhejiang Green Pesticide Collaborative Innovation Center, Zhejiang University of Technology, Hangzhou 310014, China; 2112101168@zjut.edu.cn (Z.Y.); 211122010185@zjut.edu.cn (R.H.); 2112101129@zjut.edu.cn (J.C.)

**Keywords:** diamide insecticides, indane, optically active, synthesis, insecticidal activity

## Abstract

Diamide insecticides have always been a hot research topic in the field of pesticides. To further discover new compounds with high activity and safety, indane and its analogs were introduced into chlorantraniliprole, and a battery of chlorfenil derivatives, including indane and its analogs, were designed and prepared for biological testing. Their characterization and verification were carried out through nuclear magnetic resonance (NMR) and high-resolution mass spectrometry (HRMS). Biological detection showed that all the compounds exhibited good insecticidal activity against *Mythimna separata.* At 0.8 mg/L, the insecticidal activity of compound **8q** against *Mythimna separata* was 80%, which was slightly better than that of chlorantraniliprole. The results of the structure–activity relationship (SAR) analysis indicated that the indane moiety had a significant effect on insecticidal activity, especially in the R-configuration. The results indicated that chlorantraniliprole derivatives containing indane groups could serve as pilot compounds for the further development of new insecticides.

## 1. Introduction

Pesticides are irreplaceable in effectively preventing and controlling diseases and pests and ensuring food security. Chemical control via simple methods and with quick results is an important means of controlling agricultural pests [1,2,3]. Chemical insecticides are broadly utilized in farming to control pests and improve the quality and yield of agricultural products. When harmful organisms emerge, chemical control may become the only effective measure. Although chemical pesticides have made significant contributions to increasing agricultural production, they also have many drawbacks. The extensive use of pesticides, while harmful agricultural organisms are controlled, also causes environmental pollution, excessive pesticide residues in agricultural products, pesticide damage, the poisoning of humans and animals, and damage to agricultural ecological balance and biodiversity. In addition, the frequent use or abuse of insecticides has led to the rapid growth of insect resistance and cross-resistance to commercially available insecticides, posing new challenges for insect prevention and management. Thus, novel insecticides with environmentally friendly, safe, efficient, and unique mechanisms of action are urgently needed to protect crops [4,5,6].

In the past few decades, diamide insecticides have attracted widespread attention as the latest major type of insecticide, ushering in a new era of synthetic insecticides [7,8]. Diamide pesticides are the most popular commercial products for controlling insects after neonicotinoid pesticides, and they have novel structures, unique mechanisms of action, high efficiency, and a broad spectrum of action. In 1988, a Japanese pesticide company reported the first ryanodine receptor (RyR) regulator insecticide, flubendiamide, and DuPont subsequently made significant modifications to the structure and functional groups of these insecticides and ultimately developed chlorantraniliprole and cyantraniliprole (Figure 1), both of which were highly successful. These compounds act on RyR in insects and thus have great potential for use in integrated pest management strategies [9,10,11,12,13,14].

RyRs are the most important type of calcium ion release channel and are known for their ability to bind with high-affinity ryanodine. Diamide insecticides are muscle toxins that inhibit the regulation of RyR and calcium ions by acting on calcium channels. An increase in the calcium ion concentration can cause muscle contraction. To control the efflux of calcium ions from the cell, they can be stored in specific locations within the cell. RyRs have this storage function and are responsible for regulating the release of calcium ions from intracellular calcium channels. Insecticides first bind to RyRs, causing the channel to be fixed in an open shape to release the calcium ions stored in the small cell body, leading to an increase in the calcium ion concentration in insect muscle tissues. Immediately, calcium ions bind to troponin, triggering a contraction reaction between actin and myosin, causing muscle fibers to contract. In addition, the release of calcium ions quickly activates the calcium ion pump, irreversibly opening up the calcium ion channels. A large amount of calcium ions are continuously released, stimulating the continuous contraction of insect muscles. Insects stop moving and feeding, and then vomit, starve, dehydrate, defecate, and eventually stiffen and die [15,16,17,18,19,20].

Indane and its analogs are common in modern pharmaceutical chemistry; they exhibit excellent structures and physicochemical properties and have been proven to be beneficial for molecular optimization. Ternary or quaternary aliphatic rings are widely used in pharmaceutical chemistry due to their ideal structural and physicochemical properties, such as their small molecular size, rigid structure, and three-dimensional framework with atomic economy. Indane groups contain aromatic benzene rings, which fuse with aliphatic cyclopentanes, providing a rigid bicyclic framework rich in various chemical properties [21,22]. Indane and its analogs have not only greatly promoted the development of medicine but also achieved great success in the field of pesticides. Several pesticides containing indane structures have been developed and have achieved excellent results. These agents include indefencarb, indazine fluconazole, fluconazole, and indefenone [23,24,25,26,27,28,29,30,31].

Based on these findings, we synthesized a batch of new compounds by introducing indane and its analogs into chlorantraniliprole. Their biological activity was evaluated, and their mode of action was also clarified. Preliminary biological tests indicated that most of the products have activity against *M. separata*. The design ideas of these products are shown in Figure 2.

## 2. Results and Discussion

### 2.1. Preparation of the Target Compounds

The preparation processes of the target compounds are shown in Figure 1. Using substituted phenylhydrazine hydrochloride as the starting material, compound **2** was synthesized by a catalytic condensation reaction with diethyl maleate in an ethanol/sodium ethanol mixture, followed by bromination, oxidation, alkaline hydrolysis, and acidification to obtain the intermediate compound **5**. Compound **5** reacts with substituted ortho aminobenzoic acid **6** in the presence of methylsulfonyl chloride to generate compound **7**. Finally, target compounds **8a**–**8t** were synthesized through the ammonolysis of indane and its analogs [32]. All the compounds were characterized by NMR and HRMS, and the results of the spectral tests and analytical data were in agreement with their presumptive structures.

### 2.2. Greenhouse Insecticidal Activity Assays

The insecticidal activities of the strains against *Mythimna separata*, *Aphis craccivora*, and *Tetranychus cinnbarinus* are shown in Table 1, with chlorantraniliprole serving as the control. Preliminary biological tests showed that most of these compounds had ideal activity against *M. separata*. Compounds **8a**, **8c**, **8d**, **8e**, **8h**, **8i**, **8j**, **8k**, **8m**, **8q**, **8r**, **8s**, **8t**, and **8u** had activities against *M. separata* of 100% at 100 mg/L. The insecticidal activities of compounds **8b**, **8f**, **8g**, **8l**, **8n**, **8o**, and **8p** against *M. separata* were also >70%. However, they had no activity against *A. craccivora* or *T. cinnbarinus*. The biological results revealed that several compounds had good biological activity against *M. separata* at a dose of 100 mg/L, and further screening experiments were subsequently carried out.

Further biological testing of the target compound against *M. separata* showed that many compounds still exhibited excellent activity against *M. separata* at low concentrations. Among them, compounds **8c** and **8q**, at 4 mg/L, had insecticidal activities against *M. separata* of 90% and 100%, respectively, which were better than that of chlorantraniliprole. Compounds **8q** and **8i,** at 0.8 mg/L, still achieved insecticidal activities of 80% and 60% against *M. separata,* respectively (Table 2).

SAR analysis revealed that the introduction of indane and its analogs could optimize the activity of the above compounds against insects. The insecticidal activity of the target compounds depended on the R_1_, R_2_, R_n_, and X atoms. However, from these data, for R_1_ and R_2_, the roles of CH_3_ and Cl in insecticidal activity were equivalent, and their advantages and disadvantages could not be compared. For the R_n_ substituent, the effect of indane was better than that of tetralin, and the position of indane had a certain influence on its insecticidal activity. Among them, the chiral structure had the greatest impact on insecticidal activity, followed by the R-configuration > racemic structure > S-configuration structure. When the pyridine pyrazole ring was replaced with a phenyl pyrazole ring, the insecticidal activity slightly decreased.

Compound **8i** exhibited significant insecticidal activity against *M. separata* (LC_50_
*=* 1.0426 mg/L), with this value being slightly lower than that of chlorantraniliprole (LC50 = 0.4674 mg/L) (Table 3).

### 2.3. Docking Analysis

To study the mode of action between the target compound and its receptor proteins, molecular docking techniques were utilized to predict the interaction mechanism between the proteins and small molecules and to determine the binding sites and modes of the template molecules and screening molecules. Compounds **8q** and chlorantraniliprole were selected for global docking with receptor proteins. The obtained modes of theoretical binding are shown in Figure 3. Chlorantraniliprole and compound **8q** were molecularly docked within the crystal structure (PDB ID: 5Y9V) of the N-terminal domain (NTD) of the diamondback moth RyR. Chlorantraniliprole and compound **8q** could bind to the target through hydrogen bonding, π–π stacking, and other means, with binding energies of −8.19 kcal/mol and −8.50 kcal/mol, respectively. These results indicated that compound **8q**, like chlorantraniliprole, also acted on RyR; however, there were certain differences in the amino acid residues and bonding between the two and the proteins. Chlorantraniliprole forms hydrogen bonds with Thr-178, Thr-194, and His-195 through two O atoms of its amide and N atoms of its pyridine. Moreover, a π–π stacking effect occurs between its pyridine ring and the imidazole ring on the side chain of His-195; on the other hand, compound **8q**’s effect is composed of O and H atoms from its amide, Cl atoms from its benzene ring, and O atoms from its amide, forming hydrogen bonds with Val-193, Thr-194, Thr-30, and His-195, respectively. Moreover, the benzene ring and pyrazole ring of compound **8q** form π–π stacking interactions with the imidazole rings of His-192 and His-195, respectively. This difference could be due to the substitution of functional groups on chlorantraniliprole to create the new compound **8q**, which led to changes in the overall structure and spatial conformation of the molecule.

## 3. Materials and Methods

### 3.1. Instrumentation

All reagents and experimental materials were acquired directly from commercial channels and utilized without any further purification. The melting points of the compounds were measured by a B-545 melting point analyzer without calibration. Silica gel (100–200 mesh) was used to separate the compounds, and NMR spectra were obtained with a Bruker AV-400/500 MHz spectrometer using DMSO-*d_6_* as the solvent. Mass spectrometry analysis was conducted on an Agilent 6545 Q-TOF liquid chromatography–mass spectrometer.

### 3.2. Synthesis

The synthesis method for intermediate compound **5** in this study is shown in Figure 1, where compounds **5a**–**5r** were obtained from commercial sources and were used directly without purification.

#### 3.2.1. General Synthesis of Compound **2**

A total of 60 mL of anhydrous ethanol was added to a three-necked reaction flask, followed by the addition of sodium ethanol (84 mmol). After thorough stirring and dissolution, compound **1** (28 mmol) was poured into the solution and slowly heated until reflux occurred. After 30 min, 33.6 mmol of diethyl maleate was slowly added, and the mixture was refluxed for more than 1 h. The reaction progress was tracked by TLC. After the reaction was completed, the mixture was cooled to 50 °C, after which the reaction mixture was neutralized with 56 mmol of glacial acetic acid. The solvent was removed under vacuum and reduced pressure, the mixture was diluted with 100 mL of water, and its aqueous phase was extracted three times using 30 mL of ethyl acetate. The organic phase was washed separately with saturated NaHCO_3_ solution and brine, anhydrous MgSO_4_ was added to dry the mixture, and the solvent was removed under reduced pressure. The obtained crude product was purified by silica gel column chromatography using ethyl acetate (EA) and petroleum ether (PE) (V_EA_:V_PE_ = 1:3) to obtain compound **2**.

#### 3.2.2. General Synthesis of Compound **3**

Compound **2** was added to acetonitrile (30 mL), and the mixture was allowed to cool to 0 °C. Then, 13.2 mmol of triethylamine and 13.2 mmol of phosphorus oxybromide were slowly added in sequence. The mixture was slowly heated to 60 °C. After 30 min, the reaction was complete, and the solvent was subsequently removed. The residue was diluted with 50 mL of saturated NaHCO_3_ solution, the aqueous phase was extracted three times using 30 mL of ethyl acetate, and the organic phases were combined and dried over anhydrous Na_2_SO_4_. The solvent was removed under reduced pressure to obtain a relatively pure sample, which did not require purification and could proceed directly to the next step.

#### 3.2.3. General Synthesis of Compound **4**

A total of 10 mmol of the previous compound **3** and 30 mL of acetone were added to a three-necked flask. When the temperature of the reaction mixture decreased to 0 °C, potassium permanganate (40 mmol) was slowly added, and the mixture was subsequently heated to room temperature. After the reaction was completed, the filter cake was filtered and washed with acetone multiple times. The solvent was removed under reduced pressure. The crude product was purified by silica gel column chromatography using EA and PE (V_EA_:V_PE_ = 1:10) to obtain pure compound **4**.

#### 3.2.4. General Synthesis of Compound **5**

Compound **4** (3 mmol) was added to a reaction bottle and dissolved in a 10 mL ethanol solution. We slowly added 1.5 mmol of sodium hydroxide aqueous solution (and 10 mL water) to the reaction system and stirred for 2 h at room temperature, reduced the pressure to remove methanol after the reaction, added 30 mL water, adjusted the system pH to 2–3 with concentrated hydrochloric acid, and generated a large amount of solid precipitates. The precipitate was collected by filtration and dried to obtain compound **5**. The mixture was directly used for subsequent procedures without further purification.

#### 3.2.5. General Synthesis of Compound **7**

Compound **5** (3 mmol) and compound **6** (3.1 mmol) were added to 20 mL of acetonitrile in a 50 mL three-necked round bottom flask. After the temperature decreased to 10 °C, 15.6 mmol of 3-methylpyridine was added dropwise. After 30 min, a solution of acetonitrile-containing methylsulfonyl chloride (7.2 mmol) was added. The temperature was held below 10 °C for 30 min, after which the mixture was allowed to rise to room temperature for 30 min to stop the reaction. An appropriate amount of concentrated HCl was added to remove pyridine, water was added, and the mixture was filtered and dried to obtain compound **7**.

#### 3.2.6. General Synthetic Procedure for Compounds **8a**–**8u**

Compound **7** (0.7 mmol) was added to acetonitrile (10 mL). During the stirring process, indane and its analogs (0.84 mmol) were added. After refluxing for 4 h, solid precipitates were observed after cooling to room temperature. The filter cake was filtered, collected, and allowed to dry naturally to obtain the target compounds **8a**–**8u**.

3-bromo-1-(3-chloropyridin-2-yl)-N-(2-((2,3-dihydro-1H-inden-1-yl)carbamoyl)-4,6-dimethylphenyl)-1H-pyrazole-5-carboxamide (8a)—white solid, yield 67.5%, m.p. 250.4–251.4 °C, HPLC 99.6%, ^1^H NMR (400 MHz, DMSO-*d_6_*) δ: 10.16 (s, 1H), 8.50 (dd, J = 4.7, 1.4 Hz, 1H), 8.42 (d, J = 8.3 Hz, 1H), 8.16 (dd, J = 8.1, 1.4 Hz, 1H), 7.61 (dd, J = 8.1, 4.7 Hz, 1H), 7.37 (s, 1H), 7.23 (t, J = 6.4 Hz, 1H), 7.20–7.16 (m, 3H), 7.15 (s, 1H), 7.10 (t, J = 7.1 Hz, 1H), 5.40 (q, J = 8.0 Hz, 1H), 2.92 (ddd, J = 15.5, 8.7, 3.0 Hz, 1H), 2.87–2.73 (m, 1H), 2.41–2.31 (m, 1H), 2.29 (s, 3H), 2.14 (s, 3H), 1.87 (dq, J = 12.5, 8.7 Hz, 1H). ^13^C NMR (101 MHz, DMSO-*d_6_*) δ: 167.53, 155.99, 149.06, 147.54, 144.34, 143.30, 140.01, 139.62, 136.64, 136.35, 134.79, 132.75, 130.12, 128.41, 127.74, 127.25, 127.00, 126.92, 126.70, 124.82, 124.57, 110.93, 54.39, 32.93, 30.23, 20.89, 18.30. HRMS (ESI): calculated for C_27_H_23_BrClN_5_O_2_Na [M + Na]^+^ 586.0619, found 586.0621.

3-bromo-N-(4-chloro-2-((2,3-dihydro-1H-inden-1-yl)carbamoyl)-6-methylphenyl)-1-(3-chloropyridin-2-yl)-1H-pyrazole-5-carboxamide (8b)—white solid, yield 62.0%, m.p. 131.4–132.2 °C, HPLC 96.2%, ^1^H NMR (400 MHz, DMSO-*d_6_*) δ: 10.22 (s, 1H), 8.67 (s, 1H), 8.50 (dd, J = 4.7, 1.4 Hz, 1H), 8.16 (dd, J = 8.1, 1.4 Hz, 1H), 7.69–7.57 (m, 1H), 7.47 (t, J = 7.8 Hz, 1H), 7.39 (d, J = 2.2 Hz, 1H), 7.37 (s, 1H), 7.23 (t, J = 5.5 Hz, 1H), 7.19 (d, J = 10.2 Hz, 1H), 7.15 (s, 1H), 7.10 (t, J = 7.1 Hz, 1H), 5.36 (q, J = 7.9 Hz, 1H), 2.92 (ddd, J = 15.5, 8.7, 3.1 Hz, 1H), 2.85–2.74 (m, 1H), 2.41–2.29 (m, 1H), 2.19 (s, 3H), 1.85 (dq, J = 12.6, 8.6 Hz, 1H). ^13^C NMR (101 MHz, DMSO-*d_6_*) δ: 166.02, 155.96, 149.01, 147.56, 144.08, 143.35, 139.92, 139.65, 139.13, 136.65, 136.39, 131.63, 130.10, 128.40, 127.80, 127.25, 127.03, 126.72, 126.12, 124.82, 124.65, 111.08, 54.47, 32.95, 30.23, 18.17. HRMS (ESI): calculated for C_26_H_20_BrCl_2_N_5_O_2_Na [M + Na]^+^ 606.0059, found 606.0070.

3-bromo-1-(3-chloropyridin-2-yl)-N-(2,4-dichloro-6-((2,3-dihydro-1H-inden-1-yl)carbamoyl)phenyl)-1H-pyrazole-5-carboxamide (8c)—white solid, yield 68.4%, m.p. 177.8–178.1 °C, HPLC 92.8%, ^1^H NMR (400 MHz, DMSO-*d_6_*) δ: 10.51 (s, 1H), 8.72 (d, J = 7.9 Hz, 1H), 8.50 (dd, J = 4.7, 1.4 Hz, 1H), 8.16 (dt, J = 7.2, 3.6 Hz, 1H), 7.86 (d, J = 2.2 Hz, 1H), 7.64–7.61 (m, 1H), 7.53 (d, J = 2.3 Hz, 1H), 7.46 (s, 1H), 7.22 (dd, J = 10.9, 4.4 Hz, 2H), 7.16 (d, J = 7.1 Hz, 1H), 7.10 (t, J = 6.3 Hz, 1H), 5.36–5.29 (m, 1H), 2.93–2.87 (m, 1H), 2.82 (t, J = 8.2 Hz, 1H), 2.33 (ddd, J = 11.4, 7.9, 3.5 Hz, 1H), 1.81 (ddd, J = 16.9, 10.5, 6.4 Hz, 1H). ^13^C NMR (101 MHz, DMSO-*d_6_*) δ: 164.87, 155.96, 148.99, 147.58, 143.84, 143.39, 139.64, 139.16, 138.53, 133.94, 132.49, 130.87, 130.68, 128.54, 127.86, 127.66, 127.30, 127.11, 126.74, 124.83, 124.71, 111.43, 54.57, 33.0, 30.23. HRMS (ESI): calculated for C_25_H_17_BrCl_3_N_5_O_2_Na [M + Na]^+^ 625.9526, found 625.9523.

(S)-3-bromo-1-(3-chloropyridin-2-yl)-N-(2-((2,3-dihydro-1H-inden-1-yl)carbamoyl)-4,6-dimethylphenyl)-1H-pyrazole-5-carboxamide (8d)—white solid, yield 69.2%, m.p. 151.5–151.8 °C, HPLC 99.3%, ^1^H NMR (400 MHz, DMSO-*d_6_*) δ: 10.15 (s, 1H), 8.54–8.48 (m, 1H), 8.42 (d, J = 8.1 Hz, 1H), 8.20–8.11 (m, 1H), 7.61 (dd, J = 8.1, 4.7 Hz, 1H), 7.36 (s, 1H), 7.23 (t, J = 6.6 Hz, 1H), 7.18 (t, J = 5.3 Hz, 3H), 7.14 (s, 1H), 7.09 (t, J = 7.1 Hz, 1H), 5.39 (q, J = 8.0 Hz, 1H), 2.92 (ddd, J = 15.2, 8.6, 2.8 Hz, 1H), 2.86–2.75 (m, 1H), 2.40–2.31 (m, 1H), 2.28 (s, 3H), 2.14 (s, 3H), 1.86 (dq, J = 12.4, 8.7 Hz, 1H). ^13^C NMR (101 MHz, DMSO-*d_6_*) δ: 167.51, 155.98, 149.06, 147.54, 144.34, 143.29, 140.01, 139.62, 136.62, 136.34, 134.79, 132.74, 130.12, 128.40, 127.73, 127.24, 127.00, 126.92, 126.70, 124.82, 124.56, 110.92, 54.38, 32.92, 30.22, 20.88, 18.29. HRMS (ESI): calculated for C_27_H_23_BrClN_5_O_2_Na [M + Na]^+^ 586.0610, found 586.0616.

(S)-3-bromo-N-(4-chloro-2-((2,3-dihydro-1H-inden-1-yl)carbamoyl)-6-methylphenyl)-1-(3-chloropyridin-2-yl)-1H-pyrazole-5-carboxamide (8e)—white solid, yield 62.0%, m.p. 160.4–160.7 °C, HPLC 96.7%, ^1^H NMR (400 MHz, DMSO-*d_6_*) δ: 10.26 (s, 1H), 8.65 (d, J = 6.3 Hz, 1H), 8.50 (d, J = 3.9 Hz, 1H), 8.15 (t, J = 12.6 Hz, 1H), 7.70–7.55 (m, 1H), 7.48 (t, J = 9.0 Hz, 1H), 7.38 (s, 2H), 7.22 (dd, J = 11.4, 7.2 Hz, 2H), 7.18–7.14 (m, 1H), 7.10 (t, J = 7.1 Hz, 1H), 5.36 (q, J = 7.9 Hz, 1H), 2.92 (ddd, J = 15.3, 8.7, 3.0 Hz, 1H), 2.85–2.75 (m, 1H), 2.40–2.30 (m, 1H), 2.19 (s, 3H), 1.85 (dq, J = 12.6, 8.6 Hz, 1H). ^13^C NMR (101 MHz, DMSO-*d_6_*) δ: 166.01, 155.95, 148.99, 147.57, 144.08, 143.35, 139.78, 139.66, 139.18, 136.61, 136.53, 131.64, 131.38, 128.39, 127.80, 127.26, 127.05, 126.72, 126.12, 124.82, 124.65, 111.11, 54.47, 32.94, 30.22, 18.15. HRMS (ESI): calculated for C_26_H_20_BrCl_2_N_5_O_2_Na [M + Na]^+^ 606.0072, found 606.0070.

(S)-3-bromo-1-(3-chloropyridin-2-yl)-N-(2,4-dichloro-6-((2,3-dihydro-1H-inden-1-yl)carbamoyl)phenyl)-1H-pyrazole-5-carboxamide (8f)—yellow solid, yield 71.1%, m.p. 218.7–219.1 °C, HPLC 97.6%, ^1^H NMR (400 MHz, DMSO-*d_6_*) δ: 10.51 (s, 1H), 8.72 (d, J = 7.9 Hz, 1H), 8.50 (d, J = 4.3 Hz, 1H), 8.16 (d, J = 8.0 Hz, 1H), 7.86 (s, 1H), 7.62 (dd, J = 8.0, 4.7 Hz, 1H), 7.53 (d, J = 1.7 Hz, 1H), 7.46 (s, 1H), 7.22 (t, J = 8.7 Hz, 2H), 7.15 (d, JZ = 7.3 Hz, 1H), 7.10 (t, J = 7.3 Hz, 1H), 5.32 (q, J = 8.0 Hz, 1H), 2.96–2.86 (m, 1H), 2.85–2.74 (m, 1H), 2.39–2.29 (m, 1H), 1.82 (dq, J = 12.5, 8.5 Hz, 1H). ^13^C NMR (101 MHz, DMSO-*d_6_*) δ: 164.88, 155.97, 149.01, 147.56, 143.84, 143.39, 139.63, 139.24, 138.43, 133.91, 132.39, 131.06,130.68, 128.55, 127.86, 127.67, 127.27, 127.10, 126.74, 124.83, 124.71, 111.40, 54.57, 33.01, 30.23. HRMS (ESI): calculated for C_25_H_17_BrCl_3_N_5_O_2_Na [M + Na]^+^ 625.9520, found 625.9523.

3-bromo-1-(3-chloropyridin-2-yl)-N-(2-((2,3-dihydro-1H-inden-2-yl)carbamoyl)-4,6-dimethylphenyl)-1H-pyrazole-5-carboxamide (8g)—brown solid, yield 65.4%, m.p. 252.5–252.8 °C, HPLC 98.2%, ^1^H NMR (400 MHz, DMSO-*d_6_*) δ: 10.10 (s, 1H), 8.49 (dd, J = 4.7, 1.4 Hz, 1H), 8.37 (d, J = 7.1 Hz, 1H), 8.17 (dd, J = 8.1, 1.4 Hz, 1H), 7.61 (dd, J = 8.1, 4.7 Hz, 1H), 7.39 (s, 1H), 7.20 (dt, J = 7.2, 3.6 Hz, 2H), 7.15 (dd, J = 5.7, 3.3 Hz, 3H), 7.13 (s, 1H), 4.62–4.49 (m, 1H), 3.15 (dd, J = 16.0, 7.8 Hz, 2H), 2.80 (dd, J = 16.0, 6.4 Hz, 2H), 2.28 (s, 3H), 2.13 (s, 3H). ^13^C NMR (101 MHz, DMSO-*d_6_*) δ: 167.61, 155.89, 149.05, 147.52, 141.70 (2C), 139.99, 139.62, 136.53, 136.15, 134.79, 132.65, 130.08, 128.43, 127.26, 127.03, 126.84, 126.81 (2C), 124.87 (2C), 110.86, 50.68, 39.26 (2C), 20.86, 18.27. HRMS (ESI): calculated for C_27_H_23_BrClN_5_O_2_Na [M + Na]^+^ 586.0602, found 586.0616.

3-bromo-N-(4-chloro-2-((2,3-dihydro-1H-inden-2-yl)carbamoyl)-6-methylphenyl)-1-(3-chloropyridin-2-yl)-1H-pyrazole-5-carboxamide (8h)—yellow solid, yield 62.0%, m.p. 226.8–227.1 °C, HPLC 98.4%, ^1^H NMR (400 MHz, DMSO-*d_6_*) δ: 10.22 (s, 1H), 8.58 (d, J = 7.0 Hz, 1H), 8.50 (dd, J = 4.6, 1.1 Hz, 1H), 8.18 (dd, J = 8.0, 1.1 Hz, 1H), 7.62 (dd, J = 8.1, 4.7 Hz, 1H), 7.47 (d, J = 1.9 Hz, 1H), 7.40 (s, 1H), 7.34 (d, J = 2.1 Hz, 1H), 7.23–7.18 (m, 2H), 7.17–7.13 (m, 2H), 4.60–4.41 (m, 1H), 3.14 (dd, J = 16.1, 7.8 Hz, 2H), 2.80 (dd, J = 16.1, 6.1 Hz, 2H), 2.18 (s, 3H). ^13^C NMR (101 MHz, DMSO-*d_6_*) δ: 166.08, 155.86, 148.98, 147.55, 141.66 (2C), 139.67, 139.00, 136.58, 131.94, 131.82, 131.57, 131.30, 128.42, 127.28, 127.08, 126.81 (2C), 126.08, 124.88 (2C), 111.07, 50.72, 39.21 (2C), 18.12. HRMS (ESI): calculated for C_26_H_20_BrCl_2_N_5_O_2_Na [M + Na]^+^ 606.0095, found 606.0070.

3-bromo-1-(3-chloropyridin-2-yl)-N-(2,4-dichloro-6-((2,3-dihydro-1H-inden-2-yl)carbamoyl)phenyl)-1H-pyrazole-5-carboxamide (8i)—white solid, yield 68.4%, m. p. 201.3–201.6 °C, HPLC 95.2%, ^1^H NMR (400 MHz, DMSO-*d_6_*) δ: 10.48 (s, 1H), 8.68 (d, J = 7.1 Hz, 1H), 8.50 (dd, J = 4.6, 1.2 Hz, 1H), 8.19 (dd, J = 8.1, 1.2 Hz, 1H), 7.85 (t, J = 6.1 Hz, 1H), 7.63 (dd, J = 8.1, 4.7 Hz, 1H), 7.49 (d, J = 2.3 Hz, 1H), 7.46 (s, 1H), 7.24–7.18 (m, 2H), 7.17–7.12 (m, 2H), 4.59–4.42 (m, 1H), 3.14 (dd, J = 16.1, 7.8 Hz, 2H), 2.77 (dd, J = 16.1, 6.0 Hz, 2H). ^13^C NMR (101 MHz, DMSO-*d_6_*) δ: 164.90, 155.86, 148.98, 147.56, 141.60 (2C), 139.65, 139.11, 138.44, 133.70, 132.37, 130.83, 130.60, 128.55, 127.64, 127.27, 127.15, 126.83 (2C), 124.89 (2C), 111.38, 50.73, 39.19 (2C). HRMS (ESI): calculated for C_25_H_17_BrCl_3_N_5_O_2_Na [M + Na]^+^ 625.9511, found 625.9523.

3-bromo-1-(3-chloropyridin-2-yl)-N-(2,4-dimethyl-6-((1,2,3,4-tetrahydronaphthalen-1-yl)carbamoyl)phenyl)-1H-pyrazole-5-carboxamide (8j)—brown solid, yield 45.0%, m.p. 142.7–143.3 °C, HPLC 99.7%, ^1^H NMR (400 MHz, DMSO-*d_6_*) δ: 10.14 (s, 1H), 8.49 (d, J = 4.0 Hz, 1H), 8.35 (d, J = 8.4 Hz, 1H), 8.16 (d, J = 7.6 Hz, 1H), 7.61 (dd, J = 8.0, 4.7 Hz, 1H), 7.36 (s, 1H), 7.17 (s, 2H), 7.13 (t, J = 6.4 Hz, 2H), 7.07 (d, J = 7.2 Hz, 1H), 7.02 (t, J = 7.2 Hz, 1H), 5.05 (s, 1H), 2.70 (d, J = 18.6 Hz, 2H), 2.28 (s, 3H), 2.14 (s, 3H), 1.86 (d, J = 5.1 Hz, 2H), 1.70 (d, J = 5.3 Hz, 2H). ^13^C NMR (101 MHz, DMSO-*d_6_*) δ: 167.18, 155.93, 149.08, 147.53, 139.95, 139.61, 137.79, 137.47, 136.57, 136.22, 134.82, 132.71, 130.03, 129.06, 128.57, 128.44, 127.24, 126.98 (2C), 126.95, 126.21, 110.89, 47.29, 30.08, 29.29, 20.89, 20.67, 18.29. HRMS (ESI): calculated for C_28_H_25_BrClN_5_O_2_Na [M + Na]^+^ 600.0780, found 600.0772.

3-bromo-N-(4-chloro-2-methyl-6-((1,2,3,4-tetrahydronaphthalen-1-yl)carbamoyl)phenyl)-1-(3-chloropyridin-2-yl)-1H-pyrazole-5-carboxamide (8k)—pink solid, yield 60.0%, m.p. 199.3–199.6 °C, HPLC 98.6%, ^1^H NMR (400 MHz, DMSO-*d_6_*) δ: 10.28 (s, 1H), 8.63 (t, J = 15.8 Hz, 1H), 8.51 (d, J = 4.6 Hz, 1H), 8.16 (d, J = 8.0 Hz, 1H), 7.62 (dd, J = 8.0, 4.7 Hz, 1H), 7.49 (d, J = 1.7 Hz, 1H), 7.40 (s, 1H), 7.36 (d, J = 1.9 Hz, 1H), 7.18–7.11 (m, 2H), 7.08 (d, J = 7.3 Hz, 1H), 7.04 (t, J = 7.3 Hz, 1H), 5.03 (d, J = 3.9 Hz, 1H), 2.70 (d, J = 19.9 Hz, 2H), 2.20 (s, 3H), 1.87 (d, J = 5.2 Hz, 2H), 1.71 (t, J = 9.9 Hz, 2H). ^13^C NMR (101 MHz, DMSO-*d_6_*) δ: 165.73, 155.90, 149.03, 147.55, 139.67, 139.65, 139.04, 137.51 (2C), 136.64, 131.79, 131.59, 131.33, 129.08, 128.71, 128.45, 127.23, 127.08, 127.07, 126.24, 126.16, 111.11, 47.42, 30.01, 29.30, 20.56, 18.16. HRMS (ESI): calculated for C_27_H_22_BrCl_2_N_5_O_2_Na [M + Na]^+^ 620.0222, found 620.0226.

3-bromo-1-(3-chloropyridin-2-yl)-N-(2,4-dichloro-6-((1,2,3,4-tetrahydronaphthalen-1-yl)carbamoyl)phenyl)-1H-pyrazole-5-carboxamide (8l)—yellow solid, yield 67.5%, m.p. 187.0–187.6 °C, HPLC 97.7%, ^1^H NMR (400 MHz, DMSO-*d_6_*) δ: 10.46 (s, 1H), 8.84–8.59 (m, 1H), 8.50 (dd, J = 4.6, 1.1 Hz, 1H), 8.15 (dt, J = 15.1, 7.5 Hz, 1H), 7.85 (d, J = 1.8 Hz, 1H), 7.63 (dd, J = 8.0, 4.7 Hz, 1H), 7.54–7.39 (m, 2H), 7.14 (dd, J = 12.5, 5.5 Hz, 2H), 7.08 (d, J = 7.4 Hz, 1H), 7.04 (t, J = 7.3 Hz, 1H), 4.99 (d, J = 3.7 Hz, 1H), 2.81–2.64 (m, 2H), 1.85 (d, J = 5.0 Hz, 2H), 1.66 (dd, J = 16.6, 8.9 Hz, 2H). ^13^C NMR (101 MHz, DMSO-*d_6_*) δ: 164.57, 155.89, 149.05, 147.55, 139.62, 139.16, 137.54, 137.25, 136.02, 133.68, 132.23, 130.63, 129.17, 129.12, 128.82, 128.59, 127.70, 127.26, 127.15, 127.11, 126.26, 111.37, 47.52, 29.96, 29.27, 20.44. HRMS (ESI): calculated for C_26_H_19_BrCl_3_N_5_O_2_Na [M + Na]^+^ 639.9674, found 639.9680.

(S)-3-bromo-1-(3-chloropyridin-2-yl)-N-(2,4-dimethyl-6-((1,2,3,4-tetrahydronaphthalen-1-yl)carbamoyl)phenyl)-1H-pyrazole-5-carboxamide (8m)—white solid, yield 67.3%, m.p. 147.0–147.4 °C, HPLC 99.1%, ^1^H NMR (400 MHz, DMSO-*d_6_*) δ: 10.14 (s, 1H), 8.49 (d, J = 3.7 Hz, 1H), 8.35 (d, J = 8.5 Hz, 1H), 8.16 (d, J = 8.0 Hz, 1H), 7.61 (dd, J = 8.1, 4.7 Hz, 1H), 7.35 (s, 1H), 7.17 (s, 2H), 7.13 (t, J = 6.5 Hz, 2H), 7.07 (d, J = 7.2 Hz, 1H), 7.02 (t, J = 7.3 Hz, 1H), 5.06 (d, J = 3.9 Hz, 1H), 2.70 (d, J = 19.1 Hz, 2H), 2.28 (s, 3H), 2.14 (s, 3H), 1.87 (d, J = 5.1 Hz, 2H), 1.70 (d, J = 5.4 Hz, 2H). ^13^C NMR (101 MHz, DMSO-*d_6_*) δ: 167.18, 155.94, 149.08, 147.53, 139.95, 139.61, 137.78, 137.47, 136.57, 136.22, 134.82, 132.70, 130.03, 129.06, 128.57, 128.43, 127.23, 127.01 (2C), 126.95, 126.21, 110.89, 47.29, 30.07, 29.30, 20.89, 20.67, 18.29. HRMS (ESI): calculated for C_28_H_25_BrClN_5_O_2_Na [M + Na]^+^ 600.0771, found 600.0772.

(S)-3-bromo-N-(4-chloro-2-methyl-6-((1,2,3,4-tetrahydronaphthalen-1-yl)carbamoyl)phenyl)-1-(3-chloropyridin-2-yl)-1H-pyrazole-5-carboxamide (8n)—white solid, yield 60.0%, m.p. 160.7–161.1 °C, HPLC 96.5%, ^1^H NMR (400 MHz, DMSO-d_6_) δ: 10.26 (s, 1H), 8.59 (s, 1H), 8.50 (d, J = 3.8 Hz, 1H), 8.16 (d, J = 8.0 Hz, 1H), 7.62 (dd, J = 8.0, 4.7 Hz, 1H), 7.48 (s, 1H), 7.38 (d, J = 6.2 Hz, 1H), 7.36 (s, 1H), 7.13 (d, J = 7.3 Hz, 2H), 7.08 (d, J = 7.3 Hz, 1H), 7.03 (t, J = 7.3 Hz, 1H), 5.02 (d, J = 3.7 Hz, 1H), 2.70 (d, J = 18.7 Hz, 2H), 2.19 (s, 3H), 1.86 (d, J = 5.1 Hz, 2H), 1.69 (d, J = 5.6 Hz, 2H). ^13^C NMR (101 MHz, DMSO-d_6_) δ: 165.74, 155.90, 149.06, 147.54, 139.74, 139.64, 139.01, 137.52 (2C), 136.56, 131.89, 131.59, 131.23, 129.09, 128.71, 128.46, 127.27, 127.09, 127.05, 126.26, 126.17, 111.10, 47.42, 30.02, 29.30, 20.56, 18.18. HRMS (ESI): calculated for C_27_H_22_BrCl_2_N_5_O_2_Na [M + Na]^+^ 620.0215, found 620.0226.

(S)-3-bromo-1-(3-chloropyridin-2-yl)-N-(2,4-dichloro-6-((1,2,3,4-tetrahydronaphthalen-1-yl)carbamoyl)phenyl)-1H-pyrazole-5-carboxamide (8o)—yellow solid, yield 65.0%, m.p. 192.6–193.2 °C, HPLC 95.3%, ^1^H NMR (400 MHz, DMSO-d_6_) δ: 10.45 (s, 1H), 8.91–8.62 (m, 1H), 8.49 (d, J = 4.2 Hz, 1H), 8.14 (t, J = 9.6 Hz, 1H), 7.89–7.77 (m, 1H), 7.62 (dd, J = 7.9, 4.8 Hz, 1H), 7.51 (s, 1H), 7.44 (s, 1H), 7.14 (dd, J = 13.1, 6.7 Hz, 2H), 7.10–7.06 (m, 1H), 7.02 (dd, J = 16.8, 9.5 Hz, 1H), 4.99 (s, 1H), 2.72 (s, 2H), 1.84 (d, J = 5.0 Hz, 2H), 1.74–1.60 (m, 2H). ^13^C NMR (101 MHz, DMSO-*d_6_*) δ: 164.53, 155.93, 149.15, 147.51, 139.56, 139.51, 137.52, 137.27, 136.02, 135.20, 133.54, 130.62, 129.79, 129.11, 128.85, 128.62, 127.70, 127.20, 127.16, 127.05, 126.26, 111.22, 47.50, 29.96, 29.26, 20.40. HRMS (ESI): calculated for C_26_H_19_BrCl_3_N_5_O_2_Na [M + Na]^+^ 639.9694, found 639.9680. 

(R)-3-bromo-1-(3-chloropyridin-2-yl)-N-(2,4-dimethyl-6-((1,2,3,4-tetrahydronaphthalen-1-yl)carbamoyl)phenyl)-1H-pyrazole-5-carboxamide (8p)—white solid, yield 59.7%, m.p. 149.7–150.4 °C, HPLC 98.9%, ^1^H NMR (400 MHz, DMSO-d_6_) δ: 10.14 (s, 1H), 8.49 (d, J = 4.6 Hz, 1H), 8.35 (d, J = 8.5 Hz, 1H), 8.16 (d, J = 8.0 Hz, 1H), 7.61 (dd, J = 8.0, 4.7 Hz, 1H), 7.35 (s, 1H), 7.17 (s, 2H), 7.13 (t, J = 6.6 Hz, 2H), 7.07 (d, J = 7.3 Hz, 1H), 7.02 (t, J = 7.3 Hz, 1H), 5.06 (d, J = 4.1 Hz, 1H), 2.70 (d, J = 19.1 Hz, 2H), 2.28 (s, 3H), 2.14 (s, 3H), 1.87 (d, J = 5.0 Hz, 2H), 1.78–1.59 (m, 2H). ^13^C NMR (101 MHz, DMSO-d_6_) δ: 167.18, 155.94, 149.08, 147.53, 139.95, 139.61, 137.78, 137.47, 136.57, 136.22, 134.82, 132.70, 130.03, 129.06, 128.57, 128.43, 127.23, 127.00 (2C), 126.95, 126.21, 110.89, 47.29, 30.08, 29.30, 20.89, 20.67, 18.29. HRMS (ESI): calculated for C_28_H_25_BrClN_5_O_2_Na [M + Na]^+^ 600.0758, found 600.0772.

(R)-3-bromo-N-(4-chloro-2-methyl-6-((1,2,3,4-tetrahydronaphthalen-1-yl)carbamoyl)phenyl)-1-(3-chloropyridin-2-yl)-1H-pyrazole-5-carboxamide (8q)—white solid, yield 57.2%, m.p. 161.2–161.8 °C, HPLC 97.8%, ^1^H NMR (400 MHz, DMSO-d_6_) δ: 10.26 (s, 1H), 8.59 (d, J = 7.5 Hz, 1H), 8.50 (d, J = 4.4 Hz, 1H), 8.16 (d, J = 7.9 Hz, 1H), 7.62 (dd, J = 8.0, 4.7 Hz, 1H), 7.48 (s, 1H), 7.37 (d, J = 11.5 Hz, 2H), 7.13 (d, J = 7.3 Hz, 2H), 7.08 (d, J = 7.3 Hz, 1H), 7.03 (t, J = 7.3 Hz, 1H), 5.02 (d, J = 3.6 Hz, 1H), 2.70 (d, J = 19.6 Hz, 2H), 2.20 (s, 3H), 1.86 (d, J = 5.1 Hz, 2H), 1.71 (t, J = 10.5 Hz, 2H). ^13^C NMR (101 MHz, DMSO-*d_6_*) δ: 165.71, 155.88, 149.03, 147.55, 139.71, 139.64, 139.04, 137.51 (2C), 136.64, 131.80, 131.59, 131.30, 129.08, 128.70, 128.44, 127.26, 127.08, 127.06, 126.24, 126.14, 111.08, 47.40, 30.00, 29.29, 20.55, 18.15. HRMS (ESI): calculated for C_27_H_22_BrCl_2_N_5_O_2_Na [M + Na]^+^ 620.0244, found 620.0226.

(R)-3-bromo-1-(3-chloropyridin-2-yl)-N-(2,4-dichloro-6-((1,2,3,4-tetrahydronaphthalen-1-yl)carbamoyl)phenyl)-1H-pyrazole-5-carboxamide (8r)—yellow solid, yield 49.7%, m.p. 192.9–193.2 °C, HPLC 96.2%, ^1^H NMR (400 MHz, DMSO-*d_6_*) δ: 10.45 (s, 1H), 8.90–8.59 (m, 1H), 8.49 (d, J = 4.2 Hz, 1H), 8.14 (t, J = 9.5 Hz, 1H), 7.93–7.75 (m, 1H), 7.62 (dd, J = 7.9, 4.8 Hz, 1H), 7.53 (d, J = 18.3 Hz, 1H), 7.44 (s, 1H), 7.14 (dd, J = 13.1, 6.7 Hz, 2H), 7.08 (d, J = 7.5 Hz, 1H), 7.02 (dd, J = 17.2, 9.9 Hz, 1H), 4.99 (s, 1H), 2.72 (s, 2H), 1.84 (d, J = 5.4 Hz, 2H), 1.76–1.54 (m, 2H). ^13^C NMR (101 MHz, DMSO-*d_6_*) δ: 164.53, 155.93, 149.16, 147.50, 139.56, 139.51, 137.52, 137.23, 136.02, 133.53, 132.25, 130.62, 129.78, 129.11, 128.85, 128.62, 127.70, 127.20, 127.16, 127.02, 126.26, 111.19, 47.50, 29.96, 29.26, 20.39. HRMS (ESI): calculated for C_26_H_19_BrCl_3_N_5_O_2_Na [M + Na]^+^ 639.9693, found 639.9680.

3-bromo-1-(2-chlorophenyl)-N-(2-((2,3-dihydro-1H-inden-1-yl)carbamoyl)-4,6-dimethylphenyl)-1H-pyrazole-5-carboxamide (8s)—white solid, yield 50.9%, m.p. 244.7–245.0 °C, HPLC 99.2%, ^1^H NMR (400 MHz, DMSO-*d_6_*) δ: 10.08 (s, 1H), 8.40 (d, J = 8.2 Hz, 1H), 7.57 (d, J = 7.7 Hz, 1H), 7.52–7.41 (m, 3H), 7.34 (s, 1H), 7.23 (t, J = 7.4 Hz, 1H), 7.17 (t, J = 8.7 Hz, 4H), 7.05 (t, J = 7.2 Hz, 1H), 5.39 (q, J = 7.9 Hz, 1H), 2.92 (ddd, J = 15.4, 8.7, 3.0 Hz, 1H), 2.85–2.75 (m, 1H), 2.41–2.30 (m, 1H), 2.28 (s, 3H), 2.14 (s, 3H), 1.86 (dq, J = 12.5, 8.6 Hz, 1H). ^13^C NMR (101 MHz, DMSO-*d_6_*) δ: 167.61, 156.17, 144.38, 143.33, 140.07, 138.15, 136.55, 136.32, 134.72, 132.74, 131.14, 130.86, 130.26, 129.94, 129.47, 128.19, 127.77, 126.89, 126.74, 126.61, 124.84, 124.56, 111.04, 54.37, 32.87, 30.23, 20.88, 18.32. HRMS (ESI): calculated for C_28_H_24_BrClN_4_O_2_Na [M + Na]^+^ 585.0676, found 585.0663.

3-bromo-N-(4-chloro-2-((2,3-dihydro-1H-inden-1-yl)carbamoyl)-6-methylphenyl)-1-(2-chlorophenyl)-1H-pyrazole-5-carboxamide (8t)—white solid, yield 58.6%, m.p. 256.4–256.8 °C, HPLC 96.9%, ^1^H NMR (400 MHz, DMSO-*d_6_*) δ: 10.19 (s, 1H), 8.64 (t, J = 13.1 Hz, 1H), 7.58 (d, J = 7.7 Hz, 1H), 7.50 (dd, J = 9.0, 3.4 Hz, 2H), 7.46 (s, 2H), 7.38 (s, 1H), 7.36 (s, 1H), 7.22 (dd, J = 12.9, 7.3 Hz, 2H), 7.17 (d, J = 8.3 Hz, 1H), 7.06 (t, J = 7.1 Hz, 1H), 5.36 (q, J = 7.8 Hz, 1H), 2.99–2.87 (m, 1H), 2.86–2.73 (m, 1H), 2.34 (ddd, J = 12.4, 8.0, 3.9 Hz, 1H), 2.18 (s, 3H), 1.94–1.77 (m, 1H). ^13^C NMR (101 MHz, DMSO-*d_6_*) δ: 166.10, 156.11, 144.13, 143.38, 139.17, 138.09, 132.02, 131.63, 131.31, 131.20, 130.84, 129.96, 129.50, 128.23, 127.83, 126.76, 126.67, 126.63, 126.12, 126.08, 124.84, 124.65, 111.23, 54.45, 32.86, 30.23, 18.17. HRMS (ESI): calculated for C_27_H_21_BrCl_2_N_4_O_2_Na [M + Na]^+^ 605.0125, found 605.0117.

3-bromo-1-(2-chlorophenyl)-N-(2,4-dichloro-6-((2,3-dihydro-1H-inden-1-yl)carbamoyl)phenyl)-1H-pyrazole-5-carboxamide (8u)—white solid, yield 61.1%, m.p. 254.6–255.2 °C, HPLC 96.2%, ^1^H NMR (400 MHz, DMSO-*d_6_*) δ: 10.43 (s, 1H), 8.71 (d, J = 7.7 Hz, 1H), 7.91–7.78 (m, 1H), 7.57 (d, J = 7.9 Hz, 1H), 7.53 (s, 1H), 7.49 (dd, J = 12.3, 4.0 Hz, 1H), 7.45 (s, 2H), 7.41 (s, 1H), 7.22 (dt, J = 12.2, 6.2 Hz, 2H), 7.19–7.13 (m, 1H), 7.05 (t, J = 7.1 Hz, 1H), 5.40–5.26 (m, 1H), 2.91 (dd, J = 12.2, 9.3 Hz, 1H), 2.85–2.72 (m, 1H), 2.43–2.24 (m, 1H), 1.82 (dq, J = 16.8, 8.5 Hz, 1H). ^13^C NMR (101 MHz, DMSO-*d_6_*) δ: 164.92, 156.12, 143.92, 143.41, 138.05, 133.99, 132.49, 132.16, 132.03, 131.22, 130.96, 130.64, 129.97, 129.51, 128.18, 127.93, 127.89, 127.61, 126.74, 126.67, 126.64, 124.78, 111.51, 54.55, 32.89, 30.22. HRMS (ESI): calculated for C_26_H_18_BrCl_3_N_4_O_2_Na [M + Na]^+^ 624.9560, found 624.9571.

### 3.3. Insecticidal Activity Assay

#### Determination of Insecticidal Activity in the Greenhouse

All the pests were provided by the Pesticide Creation Center of the Zhejiang Research Institute of Chemical Industry, and the activities of the target compounds **8a**–**8u** against *Mythimna separata* (*M. separata*)*, Aphis craccivora* (*A. Craccivora*), and *Tetranychus cinnbarinus* (*T. cinnbarinus*) were evaluated according to a previously reported method [33,34,35].

The test substance was dissolved in DMF with 0.1% Tween-80 emulsifier to generate a 1.0% stock solution, which was subsequently diluted with distilled water to the required dosage for the experiment.

For the leaf soaking method, the test target was *Mythimna separata* (*M. separata*). After fully soaking a proper amount of corn leaves in the prepared solution, the leaves were allowed to dry naturally in the shade. The plants were subsequently placed in culture dishes with filter paper. Ten 3rd instar middle slime worm larvae were connected per dish. The samples were placed in an observation room at 24~27 °C for culture. After 2 days of investigation, the worm body was palpated with a brush. If there was no reaction, the sample was regarded as dead.

The spray methods used were as follows: the test targets were *Tetranychus cinnbarinus* (*T. cinnbarinus*) and *Aphis craccivora* (*A. craccivora*); broad bean leaves infested with *T. cinnbarinus* and *Aphis craccivora* were placed under a Potter spray tower for spray treatment. After treatment, *T. cinnbarinus* was placed in the observation room at 24~27 °C for culture, *A. craccivora* was placed in the observation room at 20~22 °C for culture, and the results were collected 2 days later. The insects were touched with a brush and those without any response were considered dead. The results are compiled in Table 1 and Table 2.

The assessments were based on a percentage range of 0–100, where 0 meant no activity and 100 meant total kill.

### 3.4. Molecular Docking Analysis

The target protein structure was obtained from the protein database, and the target protein was the crystal structure of the NTD of the diamondback moth RyR (PDB ID: 5Y9V) [36]. Compound **8q** and chlorantraniliprole were docked with the target protein. The structures of the target proteins and small molecules were processed using AutoDockTools 4.2 software by adding Gasteiger Hucker empirical charges and H atoms, combining nonpolar hydrogen, and establishing rotatable bonds. In small-molecule structures, the partial bonds between heavy atoms were set as rotatable bonds, while template molecules were thought to have rigid structures. An optimal binding conformation was selected to determine the binding site and binding mode between the template molecule and the screened molecule (Figure 3).

### 3.5. Molecular Safety Analysis

To fully explore the other properties of synthesized compounds, we used the ADMETlab 2.0 platform to predict and evaluate the safety of synthesized compounds. Compound **8q** was selected as a representative compound, and its medicinal chemistry, absorption, metabolism, distribution, excretion, toxicity and other properties were analyzed and compared with chlorantraniliprole. The prediction results showed that it is similar to chloramphenicol and is safe for human health at the prescribed dosage. The results are shown in the Appendix A.

## 4. Conclusions

In summary, a batch of synthesized chlorantraniliprole derivatives containing indane and its analogs showed good insecticidal effects against *M. separata*, indicating that introducing indane and its analogs into diamide insecticides was an effective strategy for developing efficient insecticides. In particular, at a low concentration of 0.8 mg/L, compound **8q** exhibited greater insecticidal activity than chlorantraniliprole. The good lethality of compound **8q** against *M. separata* indicated its value as a candidate insecticide for further optimization.

## Data Availability

Samples of the compounds are not available from the authors.

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
