# Peer review of "Synthesis and Insecticidal Activity of Novel Anthranilic Diamide Insecticides Containing Indane and Its Analogs"

_ijms, 2024, doi:10.3390/ijms25042445_

Round 1
Reviewer 1 Report
Comments and Suggestions for Authors
The manuscript entitled “Synthesis and Insecticidal Activity of Novel Anthranilic Diamide Insecticides Containing Indane and its Analogs", by Yang et al describes the synthesis of a series of anthranilic diamide compounds, their biological activity and computational studies.
The work developed is relevant; the results obtained are interesting and are reasonable discussed. The manuscript is globally well written and the references are adequate. The conclusion is appropriated and shows the main results obtained.
In biological activity studies, concentration is indicated in mg/mL and not in molarity units. In the case of compounds obtained synthetically, pure compounds were certainly used, why weren't molarity units used? Is more suitable in these cases.
The aspects indicated below should also be considered when reviewing the article:
- the yields of synthesized compounds must be rounded to the nearest unit;
- in line 195, “The crude product was cleaned by silica gel column chromatography using EA and PE”, “cleaned” should be replaced by “purified”.
- in lines 202-203, and in the remaining cases in which this happens – “Filter and collect the filter cake.”, should be rewritten in a more clearer way.
After considering the aspects mentioned above, the manuscript deserves publication in International Journal of Molecular Sciences.
Comments on the Quality of English LanguageMinor editing of English language required.
Reviewer 2 Report
Comments and Suggestions for Authors
Yang et al. Reported the synthesis and insecticidal activity of novel anthranilic diamide Insecticides containing indane and its analogs. The manuscript is interesting and well-organized. I have some major concerns that need to be addressed.
1. Work in this manuscript lacks novelty; only minor modifications have been done in previously reported pharmacophore.
2. The purity of these compounds should also be checked by HPLC.
3. An EC50/ED50 value should be calculated using different compound concentrations for insecticidal activity.
4. What is the target of these compounds in insects?
5. The toxicity of these molecules should be confirmed in normal cell lines.
6. Typo and grammatical errors should be corrected.
Comments on the Quality of English LanguageMinor editing of English language required
